# A Study on the Mechanism of Environmental Information Disclosure Oriented to the Construction of Ecological Civilization in China

Bowen Lu  and Shangzhi Yue *

College of Economics and Management, Northeast Forestry University, Harbin 150040, China; lubowen@nefu.edu.cn
* Correspondence: yueshangzhi@nefu.edu.cn; Tel.: +86-156-6385-3812

**Abstract:** In the construction of ecological civilization, China has been paying attention to environmental issues, and environmental information in sustainability reports has become an essential element in the construction of ecological civilization. The study of environmental information disclosure is beneficial to the construction-of-ecological civilization system and contributes to the "double carbon" goal. This paper constructs a theoretical system based on the Pigou tax, Coase's theorem and labor value theory. By analyzing the "rights, responsibilities and benefits" of different stakeholders and defining the supply and demand for environmental information disclosure, we obtain a logical framework for environmental information disclosure in the construction of ecological civilization in China, laying a theoretical foundation for subsequent research and facilitating the prospect of carbon-information-disclosure theoretical research.

**Keywords:** ecological civilization; environmental information disclosure; theoretical analysis; supply and demand analysis; stakeholders

## 1. Introduction

The Chinese government is focused on environmentally sustainable development. In 2003, it put forward the "scientific concept of comprehensive, coordinated and sustainable development", and in 2007, China proposed, in the report of the 17th National Congress, to "build an ecological civilization". Since then, the construction of ecological civilization has begun in China. Sustainable development is the result of global efforts [1]. Therefore, the disclosure of environmental information in a sustainability report is one of the practices in the construction of ecological civilization in China.

The idea of ecological civilization has a solid ecological–philosophical foundation. The theoretical foundation of ecological civilization thought is the Marxist theory of the relationship between man and nature. Marxism's multidimensional view of the relationship between man and nature reveals the evolutionary law of the relationship between man and nature. Marx believed that "man lives by nature" [2] and that, in order not to die, man must be in a continuous process of interaction with nature, and nature is the body of man in this process. In the history of human civilization, from primitive civilization, agricultural civilization, industrial civilization to ecological civilization, ecological civilization cares for nature and requires all activities of human social production and life to incorporate natural factors, taking the harmonious coexistence and development of man and nature as the premise.

Sustainable development needs to promote the construction of ecological civilization. To build ecological civilization, we need to correctly understand and deal with the relationship between man and nature, and promote the harmonious coexistence of man and nature. These cannot be separated from the construction of an ecological civilization system. The "Opinions on Accelerating the Construction of Ecological Civilization" which promotes the

development of an ecological economy in order to establish an ecological economic system. An ecological economy mainly seeks to achieve a high degree of unity between natural and human ecology by seeking the simultaneous development of economic development and ecological construction, protecting the environment through economic innovation, caring for nature in the common development of material and spiritual civilization, and ultimately achieving a high degree of unity between natural and human ecology. In this paper, we study environmental information disclosure in sustainability reports and analyze the relationship between the supply and demand of environmental information to reflect the "one brick" of enterprises, government departments and the public in the construction of the ecological civilization system. The construction of an environmental-information-disclosure system is only a small aspect of ecological civilization-system construction, but it can drive the development of other ecological civilization systems and is the foundation of ecological civilization system construction.

In previous studies, theoretical studies on environmental information disclosure have mostly focused on one theoretical perspective for analysis, such as signaling theory [3,4], stakeholder theory [5,6], legitimacy theory [7–9], and information asymmetry theory [10]. In the study of environmental information disclosure in China, there are more empirical quantitative analysis studies and fewer theoretical studies [11]. For example, the analysis of environmental information disclosure and property rights in China [12,13], environmental information disclosure in different industries [14–16], the impact of environmental information disclosure on corporate governance [17], government coercion and media attention [18,19], etc. However, previous research lacks theoretical-system research on the different subjects of environmental information disclosure.

This paper aims to reveal the mechanism of achieving environmental information disclosure in China's ecological-civilization-construction-oriented sustainable development reports. Based on a review of the literature and data, the necessity and theoretical basis of multiple subjects of environmental information disclosure are analyzed from the perspective of economics. The necessity of environmental information disclosure in the construction of ecological civilization in China is determined by externality theories such as the Pigou tax, the Coase theorem and the labor value theory, and the stakeholders in the process of environmental information disclosure are analyzed from the perspective of "rights, responsibilities and benefits". It also analyzes the interest orientation of environmental information, methods of supply and demand analysis, and the model and logical framework of environmental information disclosure.

The innovation and contribution of this paper are mainly reflected in the following aspects. Firstly, we explain the orientation of ecological civilization construction and environmental information disclosure and analyze the inevitability of environmental information disclosure in the construction of ecological civilization; secondly, we explore the theoretical basis of environmental information disclosure oriented to ecological civilization construction from the perspective of the Pigou tax, Coase's theorem and labor value theory, and divide the stakeholders of environmental information disclosure by "power, responsibility and benefit" in management science. The study also analyzes the relationship between the supply and demand of environmental information by dividing stakeholders into "power, responsibility and benefit" from management science and "supply and demand" from economics, and constructs a framework and mechanism for the synergy of listed companies, government departments and the public. Combined with the theoretical framework constructed in this study, the foundation for future empirical research is laid. This study is an innovative study on the mechanism of environmental information disclosure oriented to the construction of ecological civilization.

This study is scientifically relevant to the analysis of the mechanism of environmental information disclosure in sustainability reports in the context of ecological civilization construction in China. This paper can improve the current theoretical system of environmental information disclosure in the context of Chinese policies and provide a theoretical reference

for the implementation and development of environmental information disclosure in line with the current environmental information disclosure in China.

This paper is divided into five parts. Section 2 elaborates the relevant theoretical basis for the following study. Section 3 identifies the subjects of environmental information disclosure in China's ecological civilization construction, Section 4 analyzes the supply and demand of environmental information in the context of China's ecological civilization construction, and, finally, environmental information disclosure in China's ecological civilization construction is discussed in Section 5.

## 2. Theoretical Basis for Environmental Information Disclosure Subjects in the Context of China's Ecological Civilization Construction

Ecological civilization as one of the important relationships in the development process of human civilization. It follows agricultural civilization and industrial civilization as the development form of human and nature. Marx pointed out that the combination of "humanism and naturalism" is a society in which man and nature are united, that is, communism, in *The Philosophical Manuscripts on Economics of 1844* [20]. Environmental information, as a public good, has more clearly defined property rights. However, companies incur transaction costs when disclosing environmental accounting information. There is an information asymmetry between shareholders and managers because of the separation of ownership and management due to principal agency. Therefore, there are externalities in environmental information disclosure. The externality of environmental information disclosure refers to the additional cost or benefit to others or to this process, when enterprises or other business entities respond to information on environmental protection, pollution prevention and abatement, and resource utilization. In addition, that cost or benefit cannot be recognized through the market, and others or society likewise cannot pay or claim through the market price. Therefore, a concerted effort by multiple parties is required to achieve the internalization of externalities.

### 2.1. The Governmental Subject in the Perspective of the Pegu Tax

The "Pigou tax" is an economic instrument to control environmental pollution, originated by Arthur Cecil Pigou in his book *The Economics of Welfare*. He became a famous British economist and is known as the "father of welfare economics". Pigou, from the perspective of welfare economics, suggests that the cause of the problem of externalities arises. This is due to the fact that Pareto optimality considers only costs and benefits from the private perspective, without taking into account that they are not the same—i.e., marginal private costs and marginal social costs, and marginal private benefits and marginal social benefits—in the process of resource allocation. Pigou argues that externalities cause market failures. The goal of internalizing externalities is achieved through government taxation of firms that cause external diseconomies, and incentives and subsidies for firms with positive externalities. The government needs to play a leading role in solving the environmental pollution problem in the "Pigou tax" proposal. The developed countries in the West have managed the environment by means of taxation and have obtained obvious results, relying on the theoretical basis of the Pigou tax, which started in the 1970s. For example, the Organization for Economic Cooperation and Development has achieved clear results on the issue of improving the quality of the ecological environment through the system of taxation. However, governments are often unable to determine the exact marginal external costs in the process of implementation due to information asymmetry, thus requiring government regulation of the environmental accounting information of enterprises.

The "Pigou tax" is a solution to the problem of environmental externalities, which requires government regulation of environmental accounting information. The reasons are as follows: On the one hand, due to information asymmetry, if the government wants to achieve the optimal level of environmental management, it needs to determine the marginal external cost and the marginal net income of enterprises, and only when the two are equal, will social welfare be maximized. However, in the process of production and

operation, enterprises will ignore environmental governance to the detriment of others and exhibit a lack of socially responsible behavior in order to pursue the maximization of corporate value, which requires the government to regulate the disclosure of corporate environmental information and ensure the authority and feasibility of government policies. On the other hand is rent-seeking behavior. In the process of government intervention, rent-seeking behavior will occur. Rent-seeking theory was proposed by the American economist Krueger and refers to the wealth-seeking transfer activities carried out by people who rely on government protection. In other words, with government intervention, companies that pollute the environment will take improper measures for their own benefit in order to seek to avoid bearing the consequences of environmental pollution and attempt to continue polluting the environment for the sake of maximizing their own interests. In order to avoid rent-seeking behavior, the government needs to carry out the regulation of the environmental information disclosure of enterprises to ensure the fairness and impartiality of the system, to avoid waste of resources and distortion of resource allocation, and to not harm the interests of the public.

*2.2. Coase's Theorem Perspective on Business Subjects*

When companies make environmental accounting disclosures, the Pigou tax is not able to explain and eliminate externalities from the perspective of the company and has limitations. Coase (1960) disagrees with Pigou's marginal thinking on externalities and suggests that "if transaction costs are zero, then output can be maximized through market transactions regardless of how initial rights are defined". Ref. [21] Coase argues that the problem of externalities is never one-way, but mutual. The attribution of environmental resource allocation to market failures is imperfect. Externalities arise from a lack of clarity over property rights, and the central argument is that the use of market mechanisms can completely solve the problem of environmental externalities. If there are no transaction costs and complete information, then the firm will reach the Pareto optimum. However, a firm cannot avoid having zero transaction costs and complete information in the production process. Under the condition of information asymmetry, environmental information disclosure by enterprises can help reduce transaction costs [22]. Theoretically, if the content of environmental information disclosures by enterprises is sufficient, enterprises will be in a market of complete information in production and operation; then, the transaction cost will tend to be zero. This realizes the state of optimal allocation of resources, as Coase said, and achieves Pareto optimality. However, in the actual trading market, complete disclosure of environmental information cannot be achieved, so how much environmental information can be disclosed by enterprises will be the key to the game of multiple parties.

*2.3. The Social Subject in the Perspective of Labor Value Theory*

Labor value is a value unique to human society, and its production confers social attributes on humans beyond the survival attributes of the species. Human beings create labor value through their labor, and value can flow with interactive behavior in society and become a source of value that further strengthens the social attributes of human beings. There is a contradictory and unified relationship between the ecological environment as natural value and labor value, the essence of which is "human own value and natural value of ecological environment" [23]. Marx pointed out in *Capital* that "labor is not the only source of the use-value, or material wealth, it produces" [20]. In the process of building ecological civilization, social subjects play an indelible role. The flow of the value of labor is created by social subjects, and, therefore, social subjects are an important part of sustainable development, are an important part in environmental information disclosure, and participate in the game between natural values and their own values in the ecological environment.

### 3. Identification of Environmental Information Disclosure Subjects Oriented to the Construction of Ecological Civilization in China

In sustainable development, the joint efforts of different stakeholders are required, and environmental information disclosure is to reveal the utilization of environmental resources and the management of environmental pollution. According to Freeman's (1984) stakeholder theory, a firm involves the interaction of managers and many stakeholders who provide the firm with relevant resources or benefit from the firm's development. Therefore, managers not only have the responsibility to meet the needs of stakeholders, but also need to balance the demands of conflicting stakeholders [24,25]. Most of the studies on stakeholders of environmental information disclosure are based on a specific stakeholder, such as governments, banks, shareholders [26], media [19], management [27], etc. In this section, based on the stakeholder analysis of environmental information disclosure, we follow the management paradigm of "power, responsibility, and benefit" to identify the multiple stakeholders of environmental information disclosure in Chinese enterprises oriented to the construction of ecological civilization.

### 3.1. Classification of Stakeholders for Environmental Information Disclosure

Stakeholders are any interested parties in an organization's external environment who are affected by the organization's decisions and actions. Since the 1980s and 1990s, with CSR as the cornerstone, stakeholder theory has developed rapidly and systematically, building its own analytical frameworks that support each other, and exploring the normative basis of CSR while seeking guidance for corporate practice. Stakeholder theory is beginning to enter the mainstream of management theory and has become the main paradigm of corporate sustainability reporting research [28].

In the process of the continuous construction of ecological civilization, Chinese environmental information disclosure stakeholders are not only studied as an overall research object. Different stakeholders have different impacts on corporate environmental information disclosure and have different dimensions in determining stakeholder classification. Freeman (1984) divides stakeholders into three dimensions: firstly, stakeholders with ownership, including managers, directors and all others who hold shares in the firm; secondly, stakeholders with economic dependencies, including managers, employees, consumers, suppliers, creditors, competitors, local communities, regulatory bodies, etc.; and thirdly, stakeholders with social interests, including government managers, special groups and the media [25]. Charwham (1992) classifies stakeholders, in terms of whether the firm has a contractual relationship or not, into contractual stakeholders (shareholders, employees, customers, distributors, suppliers, lenders) and public stakeholders (all consumers, regulators, government departments, pressure groups, media, local communities) [29]. Sirgy (2002) classifies stakeholders into internal stakeholders, external stakeholders, and distal stakeholders. The internal stakeholders include employees, managers, corporate departments and the board of directors; the external stakeholders include corporate shareholders, suppliers, creditors, local communities and the natural environment; and the distal stakeholders include competitors, consumers, public media, government agencies, voters and unions [30]. When Darnall et al. studied corporate environmental strategy, they divided corporate stakeholders into direct and indirect stakeholders [31]. In their study of stakeholders in environmental disclosure, Huang et al. divided them into external stakeholders, internal stakeholders, and intermediate stakeholder groups (environmental protection organizations, accounting firms) [32].

In the process of ecological civilization building, the stakeholders in the process of the environmental information disclosure of Chinese enterprises include: the central government, local government, public, enterprises, creditors, employees, local residents, consumers, competitors, suppliers and communities, etc. In this study, the stakeholders of corporate environmental information disclosure are divided into three dimensions: government departments, enterprises and the public.

### 3.2. Stakeholders of China's Corporate Environmental Information Disclosure Oriented to Ecological Civilization Construction

In the context of China's ecological civilization construction, corporate environmental accounting information disclosure information stakeholders are divided into three categories: First, government departments, including the central government, local governments, China Securities Regulatory Commission, China Banking Supervisory Commission and other governmental functions and institutions that supervise and manage corporate environmental information disclosure. The second is the company, including all stakeholders within the company, such as managers, creditors, shareholders, company employees, etc. The third is the public, including the public media, public associations, the public community and so on. Among them, the public media is mainly media reports and the dissemination of environmental information about the company by the self-media. The public community is, mainly, some environmental organizations, and the public community is the local residents. As shown in Figure 1.

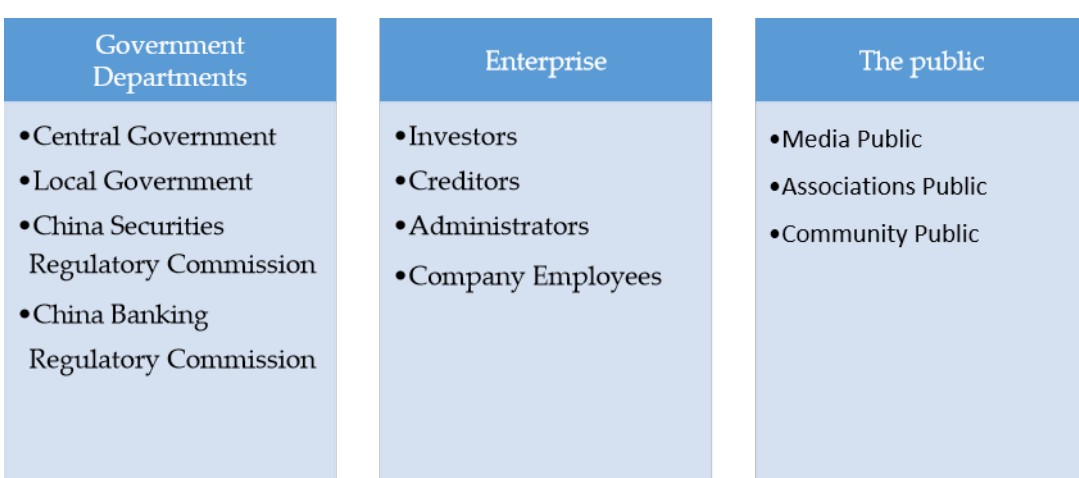

**Figure 1.** Stakeholders of environmental information disclosure in China.

This study argues that the above classification results have strong reasonableness compared with the current practice of environmental information disclosure in China. Since Chinese companies started environmental information disclosure, the central government and relevant functional departments of local governments have designated rules and assumed the responsibility of supervising companies' environmental information disclosure, while social public subjects enjoy spillover ecological services without any ecological inputs. Among these stakeholders, the micro enterprises in the market group are the main ones who perform environmental information disclosure. In addition, among the social groups, with the progress of technology level and the development of big data, the self-media and short video platforms with high traffic qualities have also become an important part of media supervision and public opinion, and the community residents are the actual experiencers of the environment and the direct beneficiaries of ecological civilization construction. Therefore, government departments, forestry-listed companies and the public will bring new opportunities from the realization of diversified subject analysis for the environmental information disclosure of forestry listed companies.

### 3.3. Analysis of the Interest Orientation of Environmental Information Disclosure Subjects in China with Ecological Civilization Orientation

In the process of corporate environmental accounting information disclosure, stakeholders of multiple subjects have different roles, positions, and orientations of responsibility interests. The utility of eco-civilization construction-oriented Chinese corporate environmental information disclosure differs for different subjects.

The government sector, as a social public institution, derives its orientation to the mechanism of environmental information disclosure for listed forestry companies from its "multiple functions and multiple responsibilities". In the process of building ecological civilization, the government includes the central government, local governments and other regulatory agencies in the disclosure of the environmental information of companies. In the process of ecological civilization construction, the central government is the initiator and overall planner of the environmental information disclosure of listed companies, and makes relevant laws and regulations, while local governments and relevant functional departments are the executors of environmental information disclosure supervision and management of listed companies. At the same time, in the process of ecological civilization construction, government departments pursue the maximization ecological benefits and need the environmental information disclosed by enterprises to grasp the existing environmental information status of the country for the ecological service guarantee mechanism, compensation mechanism, national environmental resource accounting, dual carbon strategy and many other national ecological civilization development needs in the ecological civilization construction. The responsibilities, powers and needs of government departments for the environmental information disclosure of listed companies are shown in Figure 2.

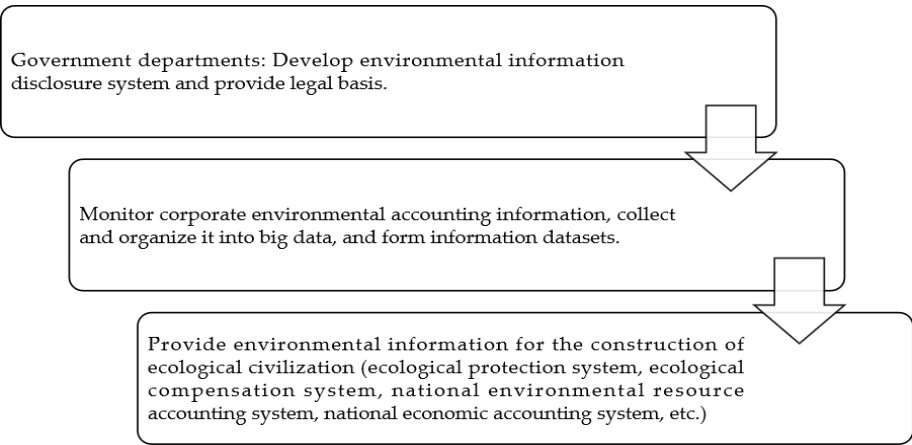

**Figure 2.** Government interest orientation in environmental information disclosure in China.

Listed companies, in the market for the purpose of profit, is a limited rational "economic man". However, in the context of China's vigorous development of ecological civilization and sustainable development, listed companies need to assume corporate social and environmental responsibility and provide environmental information in the process of environmental information disclosure; at the same time, in the process of business development, enterprises are both the demand and supply side of environmental information. Corporate governance factors exist to facilitate the disclosure of environmental accounting information [33]. The process of environmental governance within the company needs the support of environmental accounting information to further determine the company's environmental strategy, as well as environmental information needing to be subject to internal audit to provide environmental information to shareholders and creditors. However, companies, driven by economic interests, can be guilty of neglecting their environmental responsibilities and avoiding the importance of choosing not to disclose environmental information that is detrimental to their development. Therefore, companies need to strengthen their control over internal environmental management.

The public, including media, self-media, social groups, social organizations, and social residents, have multiple interests in the environmental information disclosure of listed companies. First of all, due to the rapid development of self-media, more people like to share their life around them on social platforms and be seen by more people. Some enterprises' environmental-pollution behaviors are more easily monitored by the public,

and traditional media reports add oil and vinegar, which can easily cause pressure on the enterprises polluting the environment from the aspect of public opinion, while media reports are also a double-edged sword that can enhance the social reputation of enterprises actively engaged in environmental management. Secondly, according to Maslow's hierarchical theory of needs [34], after the basic needs are maintained and satisfied, the demand for environmental services will become a new motivating factor. Therefore, social residents have the behavioral characteristics of "ecological people" and need more spiritual life functions such as recreation, entertainment, a good living environment, culture and the education of forests, which promote the physical and mental health of people and improve the social structure and spiritual civilization of human beings. Therefore, the interests of urban residents are oriented to the need for forestry-listed companies to maintain a good environment. Information disclosure guarantees the social utility of individual's living environment quality, as shown in Figure 3.

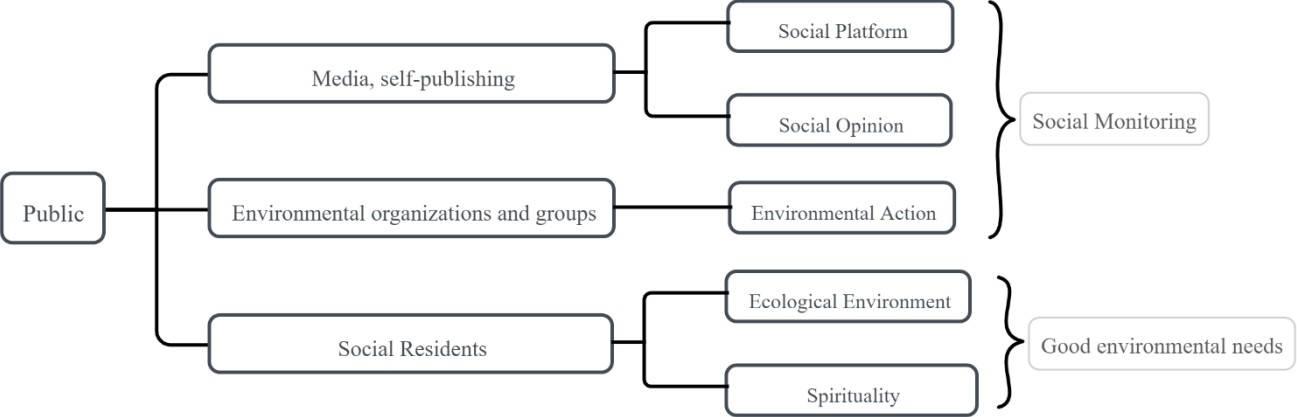

**Figure 3.** Social public interest orientation of environmental information disclosure in China.

In summary, broadly speaking, the multiple subjects of environmental information disclosure are divided into the government (both to make regulations and to be the demand side of environmental information), listed companies (the supply side of environmental information disclosure) and the public (the monitoring side of environmental information disclosure and those that enjoy the social utility of ecological civilization status). The interests of the three main parties are different, but they each have different dimensions of influence on the environmental information disclosure of the company. In the context of China's ecological civilization construction, how to disclose a company's environmental information is the result of a multi-party game and an effective way to maintain the common interests of multiple subjects.

## 4. Analysis of the Supply and Demand of Environmental Information Oriented to the Construction of Ecological Civilization in China

Stakeholders of environmental information disclosure identify multiple subjects of environmental information disclosure, including government departments, enterprises and the public, based on the interest orientation of "rights, responsibilities and benefits". In the process of environmental information disclosure, each party has its own responsibilities and needs, and the environmental information flows among them to form the environmental information disclosure system. The relationship between the supply and demand of environmental information is shown in Figure 4.

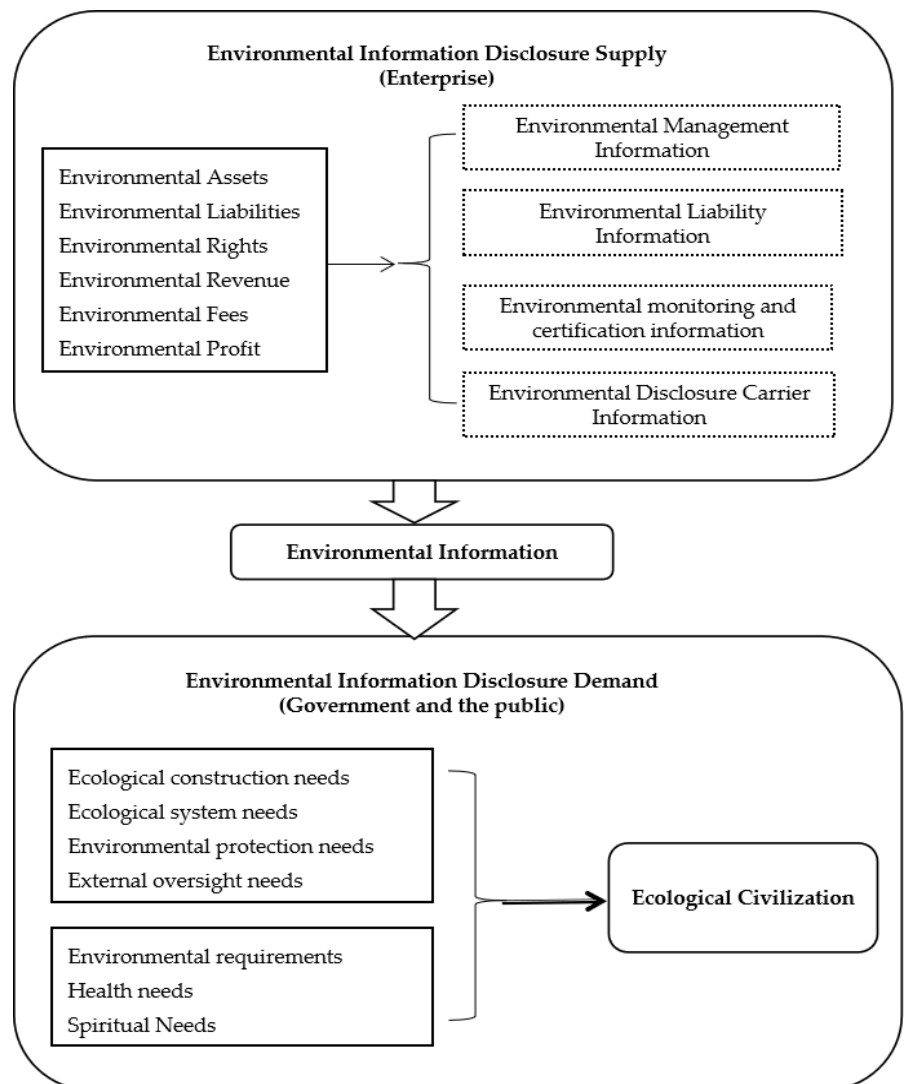

**Figure 4.** Supply and demand diagram for environmental-accounting information disclosure.

Environmental information disclosure is a manifestation of the environmental responsibility of enterprises, which is the act of providing environmental information by government departments setting up disclosure standards and is supervised by government departments and the public. At present, environmental information in the implementation of environmental information disclosure is mainly provided by listed companies, including: environmental accounting information in terms of environmental assets, environmental liabilities, environmental equity, environmental income, environmental expenses and environmental profits. At present, the main contents of the environmental accounting information disclosed by listed companies in China are: environmental management information, environmental liability information, environmental supervision and certification information, environmental carrier information, etc. Listed companies are not only the supplier of environmental information, but also the demanders of environmental information. Listed companies need to provide environmental accounting information to shareholders, creditors and other stakeholders of the enterprise to ensure that stakeholders of listed companies understand the status of the company's environmental information and maintain confidence in the management; management needs environmental information to determine the status of daily environmental management and ensure the maximization of corporate value; environmental information is needed in the process of internal auditing of the enterprise to improve the internal control of the company. The environmental

information disclosure of listed companies is a manifestation of corporate environmental responsibility.

The government sector, as the demand side of environmental information disclosure, has the following needs: (1) Ecological construction needs: China is currently in a period of the vigorous development of ecological civilization construction, which requires the collaborative governance of multiple subjects, a benign interaction between the government, enterprises, social organizations and the public in the construction of ecological civilization [35]. Environmental information, as a connection point between various subjects, is the government's need for corporate environmental information in multifaceted collaborative governance, to share information and break down information barriers. (2) Environmental protection needs: an environmental information disclosure system can play a positive role in our environment [36,37]. (3) Supervision needs: government departments need to carry out supervision functions on the environmental information of forestry-listed companies, to protect the environment on which the whole society depends, to formulate relevant policies for enterprises, to supervise the impact of enterprises on the environment through environmental information, and to punish enterprises that violate the pollution regulations.

The public is the demand side of environmental information disclosure. On the one hand, according to Maslow's hierarchy of needs theory, the first is the need for physiology and safety, and the environment as the space on which people live; a good ecological environment can provide moisture, air and safety. Events such as the Chernobyl nuclear power plant incident in the former Soviet Union in the late 1980s and the Bhopal incident in India, etc., are previous experiences of environmental pollution causing casualties. Therefore, the public needs environmental information to ensure that their environment is safe and healthy, and if there is an emergency environmental event, they can take timely measures to protect personal safety. Second is the demand for spirituality. When the public meets the need for victory, security, belonging and love, respect, and cognition, they start to pursue spiritual needs and have higher requirements and standards for the environment, or ecotourism, forest recreation, etc., or have a sense of responsibility for environmental protection, and have higher requirements for the disclosure standards of listed companies. On the other hand, the rapid development of computer technology, the advent of the 5G era and the flourishing of the self-media sector have brought a new test for monitoring the disclosure of environmental information of listed companies. Not only do the traditional media demand the environmental awareness of forestry-listed companies, but also the self-media demand the environmental awareness of listed companies. More methods of supervision are available to increase the avenues for the public's demand for environmental information.

The result of the multi-subject mechanism of environmental information disclosure depends on the result of mutual cooperation and supervision interaction among government departments, listed companies and the public in practice. The demand for environmental information from government departments is more in the pursuit of ecological benefits to achieve the overall goal of national ecological civilization construction, and the disclosure of environmental information by listed companies in compliance with national laws and regulations is both the supply of information to internal stakeholders and the supply of benefits to external stakeholders. Listed companies should not only pursue the maximization of economic interests, but should also assume corporate social responsibility in order to maximize corporate value and to form a link between the demand side and supply side of environmental information. The public demand for environmental information is more in the pursuit of social benefits, and in the supervision of forestry-listed companies environmental information disclosure at the same time, to protect their own living environment and higher life needs. Therefore, the disclosure of environmental information of listed companies requires the participation of multiple subjects, and each subject shares the corresponding responsibility and joint efforts.

## 5. Discussion

The final establishment of the eco-civilization construction-oriented corporate environmental information disclosure mechanism with the participation of multiple subjects depends on the results of mutual cooperation and interaction among government departments, listed companies and the public, in practice. Based on the theory of information asymmetry, there is information asymmetry among users of information disclosed by listed companies through environmental information, including shareholders, managers, creditors, government, and the public [9]. In the process of the production and operation activities of listed companies and economic interaction of stakeholders, there is an inequality of rights and obligations, i.e., the managers of listed companies, as the information superior party, fail to fulfill the obligation of disclosing environmental accounting information to the information inferior party (such as investors, government and public), thus causing the stakeholders to be at a disadvantage of environmental accounting information when making inappropriate judgments due to information asymmetry, which in turn causes the interests of listed companies and stakeholders to be imbalanced, and the interests of stakeholders with inferior information are ultimately harmed. Thus, to promote the environmental information disclosure system of multiple subjects, it is necessary for government departments to take the lead, formulate relevant laws and regulations, and clarify the content and methods of environmental information disclosure; for companies to play a basic role in the allocation of market resources and provide environmental information; and for the public to collaborate and supervise, forming an environmental information disclosure mechanism led by the government, with the participation of enterprises and dual supervision by the government and society. This also implies the game relationship between the government, enterprises and the public [38].

China is currently in the process of building an ecological civilization. The Chinese government's efforts to regulate environmental information disclosure continue to progress. Since the construction of ecological civilization was proposed in 2007, management policies and programs on environmental information disclosure have been continuously introduced. The policy options listed in this paper are shown in Table 1.

**Table 1.** Environmental information disclosure document.

| Year | File Name | Department |
|------|-----------|------------|
| 2008 | Environmental Information Disclosure Guidelines for Listed Companies | Shanghai Stock Exchange |
| 2010 | Environmental Information Disclosure Guidelines for Listed Companies | Environmental Information Disclosure Guidelines for Listed Companies |
| 2011 | Guidelines for the Preparation of Enterprise Environmental Reports (HJ617-2011) | Environmental Information Disclosure Guidelines for Listed Companies |
| 2017 | Cooperation Agreement on Joint Development of Environmental Information Disclosure for Listed Companies | Ministry of Environmental Protection of the People's Republic of China and China Securities Regulatory Commission |
| 2020 | Environmental information legal disclosure system reform program | Ministry of Ecology and Environment of the People's Republic of China |

At the same time, enterprises and the public are actively making efforts to disclose environmental information. The disclosure of environmental information by Chinese listed companies is gradually becoming better. Enterprises with strong industry sensitivity are gradually paying attention to their own environmental governance, actively disclosing environmental information, and building an internal environmental information disclosure system. Taking the forestry industry as an example, the overall trend of environmental information disclosure from 2015 to 2020 is on the rise. To compare, the number of com-

panies issuing ESG reports in 2020 was 1002, while the number increased to 1413 in 2021, an increase of over 40%, while the disclosure rate rose from 26.93% in 2020 to 30.27% in 2022 [39]. The public is working to monitor environmental information disclosure from media monitoring, self-media monitoring, and various environmental civil organizations. With the development of Internet technology, the self-media industry has enabled more social publics to participate in environmental information disclosure.

In order to finally reach the goal of ecological civilization, the joint efforts of the government, enterprises and society are needed. Government departments, listed companies and social residents are from three different groups, and high-quality environmental information disclosure depends on the cooperation among the three groups. Firstly, the government needs to build a perfect environmental information disclosure system at the level of laws and regulations, guide listed companies to disclose environmental information, supervise the quality of listed companies' environmental information disclosure, and avoid selective and self-serving disclosure content; on the other hand, it needs to build an environmental information sharing platform, ensure environmental information security, and solve concerns regarding listed companies' environmental information disclosure. Second, social supervision, as an auxiliary supplement to government supervision, creates public pressure or political cost on listed companies' environmental information disclosure at the social level and in terms of public opinion supervision, and creates intrinsic incentives for corporate brand reputation [40]. The government needs to screen the content of social public supervision to determine the true validity of the information. Finally, listed forestry companies disclose environmental information, accept the supervision of government departments and the public, assume corporate social responsibility, improve corporate brand reputation, and balance economic interests and environmental benefits [38].

At present, within the policy context of ecological civilization construction, most of the studies on environmental accounting information disclosure in China are empirical analyses, and although there are foundations about related theories, a clear theoretical system is lacking. The research in this paper can constitute the theoretical support of environmental information disclosure in China at present. Moreover, China has proposed the "double carbon" goal, i.e., "to achieve carbon peak by 2030 and carbon neutral by 2060", so carbon information disclosure is related to sustainable development and can be based on the theoretical system of environmental information disclosure. Therefore, this paper is a study of the theoretical system of environmental information disclosure in the context of China's current policy, and can provide a reference for the development of carbon information disclosure research and practice.

## 6. Conclusions

This paper mainly analyzes the identification of multiple subjects of environmental information disclosure, the relationship between supply and demand, and the logical relationships inherent in sustainability reports in the context of China's ecological civilization construction policy. The following conclusions are obtained: (1) In terms of environmental information disclosure subject identification, the environmental information disclosure mechanism should be based on the Pigou tax, Coase theory and labor value theory, and this paper is presented in the context of ecological civilization construction; (2) The subjects of environmental information disclosure are identified by the difference in the orientations of "rights, responsibilities and benefits", including: government departments; (3) Analysis of the supply and demand sides of environmental information of listed forestry companies from the perspective of supply and demand is presented, alongside the logical framework of multiple subjects of environmental information disclosure to lay the theoretical foundation for subsequent research.

China's sustainable development policy has been continuously improved and enhanced since its inception. The construction of ecological civilization and the "double carbon" strategy are the most recent important environmental strategies in China. Meanwhile, the improvement in the environmental information disclosure system requires the

joint efforts of the government, enterprises and the public. The three are a system, which can be studied from a certain perspective, but they should not be separated from each other, as they exist under the whole system for sub-system research. Environmental information flows among different subjects, and the behavioral orientation of the government, enterprises and the public towards environmental information disclosure will determine the presentation of environmental information disclosure. The ultimate goal is to make the optimal allocation of environmental information resources available to the government, enterprises and the public.

This paper analyzes the interest orientation, supply and demand, and logical framework of environmental information disclosure in the context of ecological civilization construction in China. However, this paper has a specific national context and has certain limitations. Whether it is suitable for the national context of other countries still needs to be explored, and whether it can be a theoretical framework for carbon information disclosure will be the direction of future research.

**Author Contributions:** Conceptualization, B.L.; methodology, B.L.; software, B.L.; validation, B.L. and S.Y.; formal analysis, B.L.; investigation, B.L.; resources, B.L.; data curation, B.L.; writing—original draft preparation, B.L.; writing—review and editing, B.L. and S.Y.; visualization, B.L.; supervision, B.L.; project administration, S.Y.; funding acquisition, S.Y. All authors have read and agreed to the published version of the manuscript.

**Funding:** This research was funded by the Humanity and Social Science Foundation of Ministry of Education of China: grant number 16YJA630072, and High-level talent research start-up fund from University of Electronic Science and Technology of China, Zhongshan Institute, grant number 420YQKN05.

**Institutional Review Board Statement:** Not applicable.

**Informed Consent Statement:** Not applicable.

**Data Availability Statement:** Not applicable.

**Conflicts of Interest:** The authors declare no conflict of interest.

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
