# Peer review of "A Study on the Mechanism of Environmental Information Disclosure Oriented to the Construction of Ecological Civilization in China"

_sustainability, doi:10.3390/su14106378_

Round 1
Reviewer 1 Report
Dear authors,
The article is of interest it presents a study on the mechanism of environmental information disclosure oriented to the construction of ecological civilization in China.
In the manuscript, the authors discuss about the environmental information in China's construction-oriented reports on the sustainable development of China's ecological civilization.
Based on the analysis of literature and data, the authors study environmental information in the construction of ecological civilization in China.
The authors use the theories of externality such as the Pigou tax, the Coase theorem and the theory of labor value.
Environmental information as well as demand and supply analysis are also analyzed.
The topic of this paper is current, interesting, and easy to read. Its structure is correct.
The article is well written, it respects the requirements for writing a scientific article.
The methodology used is adequate to achieve the objective pursued by the research.
The research results are well explained in accordance with the research carried out.
Discussions and conclusions are relevant and pertinent to the research.
The bibliography is sufficient and appropriate to the research objectives.
I haven`t any recommendations and comments else. In my opinion, the manuscript can be published if it matches the “Aims & Scope” of the journal.
Best regards,
Reviewer
Author Response
Thank you for your approval of my article! Best regards!
Reviewer 2 Report
Dear Editor,
I have completed reviewing this article (sustainability-1727714) which needs major revision.
Title: A study on the mechanism of environmental information disclosure oriented to the construction of ecological civilization in China
Suggestion: Major revision
My overall feeling towards this manuscript is positive. However, there are some major issues that need to be addressed.
- Abstract. The authors need to explore more the originality and contribution of the study in this section.
- Introduction. What are research gaps in existing literature? What’s your contributions? The authors should explicitly state in this section and then draw your contributions smoothly.
- The manuscript lacks of quantitative analysis part, please add an empirical study section.
Author Response
Please see the attachment. Thank you for your valuable comments.

Reviewer 3 Report
My impression of this study is that it summarizes the various theories well and puts together the stakeholders involved in environmental disclosure well. However, in my opinion, the issues presented in the paper are closer to the already established theories and explanations rather than any new findings. It's like reading an environmental/ESG textbook.
Also, the ones presented in this study are consistent with the research purpose (Explanation of the mechanism of environmental disclosure", lines 62-63, 84-86). However, as the title of this study suggests, elements related to the unique context of China, which is trying to build an ecological civilization, may be necessary, but I do not think that this point is well presented in the text.
Problems like these may be because the paper only defines, classifies, and declares at a high level of abstraction. In my opinion, it is necessary to present examples of various actual Chinese stakeholders responding to environmental disclosures and moving towards an ecological civilization, especially around the actions of governments, which are central to regulation and supervision. If the reader encounters only generalizations or definitions without examples, there is a limit to understanding the concepts even if the paper provides enough descriptions and summaries.
To fully achieve that, this paper may have to evolve like a case study, and it will be very different from the current version that the authors are drawing. Therefore, I agree to leave it as the subject of the subsequent study.
It seems that the specificity of the current manuscript can be achieved by including some information related to the actual actions of stakeholders in China.
Therefore, I agree to the publication of this paper on the condition that my request to "add a little explanation through real examples" regarding the actions of stakeholders in China is partially reflected in the final version.
Author Response

(The authors gave the same response as above.)

Round 2
Reviewer 2 Report
good luck.